# Optimized Protocol for Preservation of Human Platelet Samples for Fluorometric Polyphosphate Quantification

**DOI:** 10.3390/mps6040059

**Published:** 2023-06-22

**Authors:** Tomoyuki Kawase, Katsuya Suzuki, Masami Kamimura, Tomoharu Mochizuki, Takashi Ushiki

**Affiliations:** 1Division of Oral Bioengineering, Graduate School of Medical and Dental Sciences, Niigata University, Niigata 951-8514, Japan; 2Department of Transfusion Medicine, Cell Therapy and Regenerative Medicine, Niigata University Medical and Dental Hospital, Niigata 951-8520, Japan; 3Department of Orthopaedic Surgery, Graduate School of Medical and Dental Sciences, Niigata University, Niigata 951-8510, Japan; 4Division of Hematology and Oncology, Graduate School of Health Sciences, Niigata University, Niigata 951-8518, Japan; 5Department of Hematology, Endocrinology and Metabolism, Faculty of Medicine, Niigata University, Niigata 951-8510, Japan

**Keywords:** platelets, polyphosphate, preservation

## Abstract

Platelet polyphosphate (polyP) can be conveniently quantified by exploiting a recent methodological breakthrough using 4′,6-diamidino-2-phenylindole (DAPI). However, the preservation of these biological samples has not yet been standardized. In a preliminary study, potential protocols were screened, while accepted protocols were further tested in this study. Pure-platelet-rich plasma (P-PRP) samples and washed platelet suspensions were prepared using blood obtained from non-smoking healthy male donors and were fixed with ThromboFix for 20–24 h at 4 °C. Mass polyP levels were determined using a fluorometer at wavelengths of 425 and 525 nm. Platelet polyP levels were normalized to platelet counts. Statistical analyses were performed using non-parametric tests. Platelet polyP levels significantly decreased by 20% after 7 days in the platelet suspension maintained under fixed conditions at 4 °C (control). In contrast, the platelet polyP levels in both the P-PRP and washed platelet suspensions were maintained without a significant reduction for up to 6 weeks by removing ThromboFix after fixation and subsequent freezing in pure water at −80 °C. Fluorometric polyP quantification often interferes with the low specificity of DAPI binding and the wavelength used. Our validated protocols will enable long-term preservation and high-throughput polyP quantification and can be applied to relatively large cohort studies.

## 1. Introduction

Polyphosphate (polyP) is a highly anionic inorganic polymer consisting of phosphate monomers. The length and cell-associated content of biological polyP depend on the type of organism in question, for example, bacteria or mammalian cells [1]. Because of these differences and probably the simplicity of organisms compared to bacteria, the functions of polyP in mammalian cells are still poorly understood. From a practical point of view, it can also be stated that the delayed development and standardization of polyP quantification and the preservation of biological samples have hindered advances in the elucidation of polyP functions in mammalian cells.

However, recent challenges have enabled the convenient quantification of biological polyP levels using 4′,6-diamidino-2-phenylindole (DAPI) [2,3]. Although DAPI does not specifically bind to polyP [4,5,6], the right shift of the wavelength selected for fluorometric analysis increases the specificity of polyP-dependent signals in the presence of cellular compounds that bind to DAPI [2,7]. The conventional quantification method that requires complete degradation to a monomeric form and purification is time consuming and unstable for comparisons between medium and large sample sizes. However, the development of such fluorescence quantification techniques has enabled the assessment of many samples. The remaining practical issue is the preservation of the biological samples.

To overcome the latter issue, we optimized the preservation conditions for human platelets based on previously reported quantification methods [7,8,9]. Among several possible protocols, fixation, the removal of the fixative and its replacement with pure water, and preservation at −80 °C appear to be essential elements for successful preservation.

## 2. Experimental Design

Pure-platelet-rich plasma (P-PRP) was prepared from the blood obtained from five non-smoking healthy male donors (age: 24–62 years) using the double-spin method [9,10]. Platelet suspensions were prepared via centrifugation and resuspended in PBS. The basic scheme consisted of fixation (or not), replacement of the solution (or not), preservation (4 or −80 °C), and quantification. The possible combinations of individual steps are listed in Table 1.

In the preliminary study, several protocols were excluded because of practical difficulties, while several protocols were not tested because of theoretical incompatibility. Three potential protocols (in bold) were tested as promising candidates for this study.

## 3. Procedure

### 3.1. Preparation of Pure-Platelet-Rich Plasma and Platelet Suspension in PBS

The study design and consent forms for all procedures (project identification code: 2021-0126) were approved by the Ethics Committee for Human Participants at Niigata University (Niigata, Japan) and complied with the Helsinki Declaration of 1964, as revised in 2013.

Blood samples were collected from five non-smoking, healthy male donors (*n* = 6 or 7; ages 24–62 years). Despite having lifestyle-related diseases and taking medication, these donors (i.e., our team members and relatives) had no limitations in their daily activities. These donors also tested negative for HIV, HBV, HCV, or syphilis infections. A prothrombin test was performed on all blood samples using a CoaguChek^®^ XS (Roche, Basel, Switzerland), and all the results were normal. Platelet and other blood cell counts were measured using a pocH 100iV automated hematology analyzer (Sysmex, Kobe, Japan). Platelet distribution width (PDW), a conventional index for determining variation in platelet volume, and mean platelet volume (MPV), a conventional index for determining the average platelet size (volume), were also determined.

Approximately 9 mL of peripheral blood was collected in plain glass vacuum blood collection tubes (Vacutainer^®^; BD Biosciences, Franklin Lakes, NJ, USA) containing 1.5 mL of acid–citrate–dextrose solution. Whole blood samples were stored in a rotating agitator at ambient temperature (18–22 °C) until further use (<20 h). The samples were centrifuged horizontally at 415× *g* for 10 min (soft spin) (Kubota, Tokyo, Japan). The upper plasma fraction, which was ~1 mm above the interface of the plasma and red blood cell fractions, was collected and further centrifuged at 664× *g* for 3 min (hard spin) using an angle-type centrifuge (Sigma Laborzentrifugen, Osterode am Harz, Germany) to prepare P-PRP or to collect the resting platelet pellets for the preparation of the platelet suspension in PBS. Platelet counts were adjusted to 2–3.0 × 10^5^/µL.

### 3.2. Fixation and Preservation Conditions

Platelets in the form of P-PRP and PBS suspensions were fixed using ThromboFix (Beckman Coulter, Inc., Brea, CA, USA) for 20–24 h [7]. For their preservation at 4 °C, the platelets were incubated without removing the fixative. For their preservation at −80 °C, the platelets were resuspended in pure water and immediately frozen at −80 °C. In both cases, it should be noted that the pipetting of the platelet suspension should be minimized. Excessive or rough pipetting or vortexing significantly decreased platelet polyP levels.

### 3.3. Quantification of Platelet PolyP Levels

PolyP was quantified as previously described [7,9]. Briefly, fixed platelets were centrifuged at 664× *g* for 3 min, gently suspended in pure water, and incubated with DAPI (4 μg/mL) for 15 min at room temperature (18–22 °C). This mixture was directly subjected to fluorescence measurements using a fluorometer (FC-1; Tokai Optical Co., Ltd., Okazaki, Japan) with excitation and emission wavelengths of 425 and 525 nm, respectively, based on the red-shifted fluorescence of DAPI bound to polyP [2].

### 3.4. Preservation at 4 °C

As shown in Table 1, the platelets in the fixative-containing PBS were preserved at 4 °C for an initial 24 h and further preserved without replacement of the solution for up to 2 weeks. The time-course changes in the polyP, PDW, and MPV levels are shown in Figure 1. The PolyP levels gradually decreased in a time-dependent manner after cold preservation. The PolyP levels at 1 week and later were significantly lower than those in the control group at 0 h (Figure 1a). Similarly, but more drastically, the PDW and MPV levels started to decrease immediately after chilled preservation (Figure 1b,c).

### 3.5. Preservation at −80 °C

As shown in Table 1, the platelets in the fixative-containing PBS were preserved at 4 °C for an initial 24 h. At the end of the initial preservation period, the fixative-containing solution was replaced with pure water, and the platelets were counted prior to their cryopreservation at −80 °C. At each time point, the frozen platelets were thawed at room temperature and directly subjected to fluorescence quantification. The time-course changes in polyP levels are shown in Figure 2. The PolyP levels did not decrease significantly throughout the preservation period (approximately six weeks). In addition, no significant reductions were observed regardless of the initial solution type (PBS suspension or plasma suspension) used for fixation.

### 3.6. Statistical Analysis

The data are expressed as means ± SD. To compare the mean values, non-parametric analysis was performed using Friedman Repeated Measures Analysis of Variance on Ranks, followed by Dunnett’s multiple comparison method. Differences (vs. controls) were considered statistically significant at *p* < 0.05.

## 4. Expected Results and Discussion

To date, no quantification protocols have been standardized for human platelets, although DAPI has recently been used to visualize platelet polyP. From a biochemical point of view, owing to the requirements of extraction, purification, and degradation, the classic protocols developed for bacterial samples may be considered ideal and can be applied for the more accurate quantification of platelet polyP levels. However, since well-honed skills are sometimes required, these processes can also act as risk factors that reduce reproducibility.

To the best of our knowledge, rigorous comparisons have rarely been made to validate which quantification protocols are more accurate and reproducible with respect to biological samples, especially human platelets. The only exception is the study by Bru et al., which focused on yeast [11]. This study demonstrated large differences in the final quantifications between several protocols. It should be noted that, particularly in the protocol providing small values, large variations were observed, indicating less reproducibility.

The standardization of polyP quantification protocols remains a major issue in polyP research, and no well-validated quantification protocols have been standardized for human platelets. In a previous study [8], we rigorously compared fluorometric quantification data with image analysis data obtained from cytochemical visualization and suggested that our modified DAPI-based simple fluorometric protocol is reasonable but rarely counts non-specific signals in platelet samples. More importantly, we should consider how easily detachable polyP that is likely associated with the outer surface of the platelet can be preserved during preparation for better data reproducibility.

In previous studies [7,8,9], pure water was found to be the most suitable solution for platelet suspensions in relation to fluorometric polyP quantification. Furthermore, in a preliminary study, we found that platelet suspensions, including fixatives, produced insoluble compounds after thawing, which interfered with polyP quantification. Therefore, in this study, we postulated that fixative solutions should be replaced with pure water before freezing. pH-controlled buffer solutions are generally considered to be suitable for the preservation of biological samples. However, since we observed that some additives for buffering increased the fluorescence signals, we again chose pure water for preservation.

Even at −80 °C, phosphatases contained in frozen biological samples may release inorganic phosphate from polyP, thus reducing the fluorescence signal. In this study, no significant decreases in polyP levels were observed for up to 6 weeks in the frozen samples. However, a tendency toward a decrease was observed. This was probably caused by spontaneous hydrolysis and dephosphorylation via platelet-associated phosphatases. The former possibility may be prevented by lowering the preservation temperature below −80 °C. The latter can be preserved by adding appropriate phosphatase inhibitors, if any, to further prolong stable preservation.

We had limited data allowing for the discussion of the necessity of fixation. Due to the lack of fixation, an appropriate range for the linearity of the calibration curve was not obtained for the frozen and thawed samples. The thawed solution became clouded by cell nano-debris, and the linear range was always narrow. Unlike formalin, fixation with ThromboFix does not cause autofluorescence and reduces cloudiness to some extent. In addition, fixation may be beneficial for the stabilization of polyP on the plasma membrane and in cellular organelles such as dense granules. Further studies are needed to confirm this hypothesis.

In conclusion, regardless of the initial form of the platelet suspension, the protocol in which the main elements were replaced with pure water and frozen at −80 °C was validated as an optimal method for the long-term preservation of platelet samples for fluorometric polyP quantification. This protocol allows us to save time, costs, and labor by enabling the simultaneous polyP quantification of accumulated samples. Thus, this protocol can be utilized in medium- or large-cohort studies on polyP to further clarify the functions of platelet polyP.

## Figures and Tables

**Figure 1 mps-06-00059-f001:**
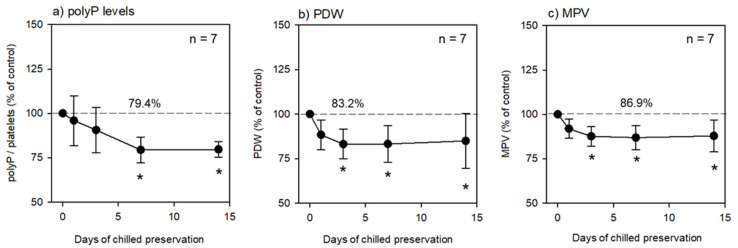
Effects of chilled preservation on (**a**) platelet polyP levels, (**b**) platelet distribution width (PDW), and (**c**) mean platelet volume (MPV). *n* = 7. Asterisks represent significant differences (*p* < 0.05) compared to the individual controls. The raw data of individual control levels were 197.3 ng/10^7^ platelets (**a**), 12.5 fL (**b**), 10.8 fL (**c**).

**Figure 2 mps-06-00059-f002:**
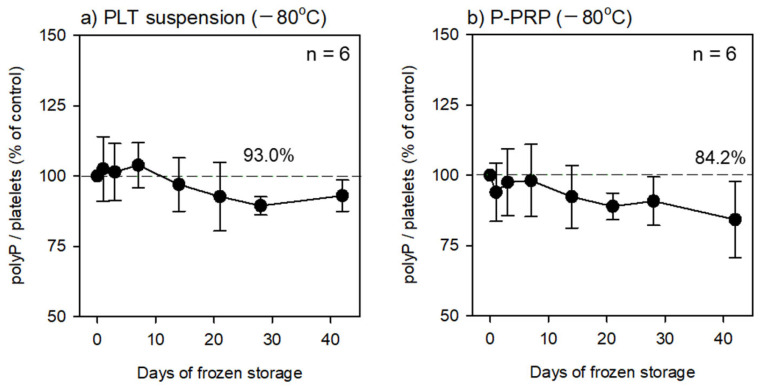
Effects of cryopreservation on platelet polyP levels in (**a**) platelet suspensions in PBS and (**b**) P-PRP. *n* = 6. Asterisks represent significant differences (*p* < 0.05) compared with the corresponding controls. The raw data of individual control levels are as follows: 168.9 (**a**) and 176.4 ng/10^7^ platelets (**b**).

**Table 1 mps-06-00059-t001:** Possible preservation protocols for polyP quantification.

Fixation (1 d, 4 °C)	Suspension	Replacement	Preservation Temperature (1 d–6 w)	To be Tested in This Study
no	Plasma	no	4 °C	Not tested *
no	PBS	no	4 °C	Not applicable **
no	Plasma	no	−80 °C	Not applicable
no	PBS	no	−80 °C	Not applicable
ThromboFix	PBS	no	4 °C	As control
ThromboFix	PBS	pure water	4 °C	Not tested
ThromboFix	PBS	no	−80 °C	Not applicable
ThromboFix	PBS	pure water	−80 °C	As a candidate
ThromboFix	Plasma	no	4 °C	Not tested
ThromboFix	Plasma	pure water	4 °C	Not tested
ThromboFix	Plasma	no	−80 °C	Not applicable
ThromboFix	Plasma	pure water	−80 °C	As a candidate

* “Not tested” indicates protocols that were not tested in the preliminary study due to theoretical reasons or for the purpose of long-term preservation. ** “Not applicable” indicates substantial decreases in polyP, conditions that are unmeasurable using cellular content or debris, or extremely narrow ranges for the measurements obtained in the preliminary study.

## Data Availability

The data are available from the corresponding author upon reasonable request.

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
