# Peer review of "Optimized Protocol for Preservation of Human Platelet Samples for Fluorometric Polyphosphate Quantification"

_mps, 2023, doi:10.3390/mps6040059_

Round 1

Reviewer 1 Report

This is a clear and straightforward study describing polyp quantification in  cryopreserved platelets. I have no concerns.

Author Response

This is a clear and straightforward study describing polyp quantification in cryopreserved platelets. I have no concerns.

Response: Thank you for your positive evaluation. We hope that the major revisions requested by other reviewers and the editor will improve the manuscript.

Reviewer 2 Report

The authors set out to examine protocols for preservation of human platelet samples for polyphosphate detection using DAPI. Platelet samples were fixed with ThromboFix (which composition is not indicated) for 24 h, at 4 oC, centrifuged, resuspended in pure water, and incubated with DAPI and their fluorescence was measured at 425-525 nm. In some cases, the platelets were kept at 4oC for two weeks or kept at -80oC for 6 weeks. They concluded that these methods allow the long-term preservation of platelet samples. The manuscript has serious flaws, and the usefulness of the methods is not evident.

Major points

1.     The method proposed to measure polyP is not adequate. Several methods have been described that could provide better estimation of polyP levels in platelets and most of these require extraction of polyP, which is not applied in this work. Incubation of platelets with DAPI at room temperature is not able to measure the real concentration of the polymer.

2.     The manuscript contains significant errors. A Table 2 is mentioned and not shown. It says that DAPI does not bind to polyP. In Line 48 it mentions degradation of the monomer (?). Figures show error bars but do not indicate their meaning. Figure values are expressed as % of control, whose values are not given. Axes of graphs should start at 0.

3.     English needs revision.

Author Response

The authors set out to examine protocols for preservation of human platelet samples for polyphosphate detection using DAPI. Platelet samples were fixed with ThromboFix (which composition is not indicated) for 24 h, at 4oC, centrifuged, resuspended in pure water, and incubated with DAPI and their fluorescence was measured at 425-525 nm. In some cases, the platelets were kept at 4oC for two weeks or kept at -80oC for 6 weeks. They concluded that these methods allow the long-term preservation of platelet samples. The manuscript has serious flaws, and the usefulness of the methods is not evident.

Response: Thank you for your comments. Please let us explain again about our purpose and the necessity of sample preservation in the polyP study. The most important fact in the background is that biological samples, such as platelets, cannot be preserved by freezing for the DAPI-based quantification of polyP levels, unlike growth factors, cytokines, and many other bioactive factors that can be quantified by the ELISA method. In this study, we emphasize that the fixation of platelets with ThromboFix, but not formalin or methanol, is necessary prior to freeze preservation. We believe that the optimized protocol would be a useful method in polyP research and developing large cohort study.

Major points

  1. The method proposed to measure polyP is not adequate. Several methods have been described that could provide better estimation of polyP levels in platelets and most of these require extraction of polyP, which is not applied in this work. Incubation of platelets with DAPI at room temperature is not able to measure the real concentration of the polymer.

Response: Thank you for your comments. We do not believe that the DAPI-based fluorometric quantification method is perfect and accurate for the determination of absolute values of polyP levels. In this quantification method, the signals can be amplified, reduced, or distorted by several factors. However, we acknowledge the advantages of this method, such as high throughput, less time consumption, and less labor wasting.

In contrast, for example, the column-based extraction method followed by colorimetric quantification of phosphate requires considerable time for fewer samples. Even if it can provide more accurate absolute values, we are also concerned about the possible non-negligible loss of polyP from the samples by several processes of extraction, purification, and degradation. In addition, colorimetric quantification is less sensitive to minor changes.

Thus, we believe that comparative data obtained from comparisons between groups are required to understand the dynamic changes in the mobilization and metabolism of polyP in platelets rather than the absolute values of polyP. We confirmed the reproducibility of the DAPI-based quantification method at room temperature and the applicability of this method for comparisons between groups as reported previously [Watanabe et al., Int J Mol Sci 22(14):7257; 2021, Ushiki et al., Physiol Rep 10(15):e15409; 2022, Int J Mol Sci 23(19):11293; 2022].

  1. The manuscript contains significant errors. A Table 2 is mentioned and not shown. It says that DAPI does not bind to polyP. In Line 48 it mentions degradation of the monomer (?). Figures show error bars but do not indicate their meaning. Figure values are expressed as % of control, whose values are not given. Axes of graphs should start at 0.

Response: Regarding “Table 2” (Line 125), it is a mistake of “Table 1.” We corrected it.

If you indicate the description “Although DAPI does not specifically bind to polyP” (Line 44), please do not ignore the keyword “specifically.” In this sentence, we mean that DAPI binding to polyP is non-specific. DAPI can bind to other polymers such as DNA; however, the wavelength for fluorometric determination increases the specificity of the signal by focusing DAPI binding to polyP. Thus, modifying the target wavelengths is important for increasing the specificity of the less specific dye.

Regarding the “degradation of the monomer”, it was caused by undesired editing. It should be “degradation to the monomer.” We corrected it.

As described in the newly added explanation in the subsection of “Statistical Analysis,” the error bar shown in the figures is the standard deviation.

Regarding 100% of each data, the control of polyP levels ranges from 168.9 to 197.3 ng/107 platelets. The control of the MPV was 10.8 fL, while the control of the PDW was 12.5 fL. The raw data have been added to the corresponding figure legends.

We aimed to determine the difference in the reduction of polyP levels among the three groups. To quickly assess the difference at a glance, we thought that the control should start at 100 instead of 0.

  1. English needs revision.

Response: We have always ordered English editing by Editage before submission. However, we have often experienced criticism, such as yours. We believe this is a human-based editing limitation and used AI-based editing software (PaperPal) in this revision. We hope this will significantly improve the quality of English.

Reviewer 3 Report

In this paper, the authors set out to provide a comparison with different DAPI-centric fluorometic quantification of polyphosphate in human platelet samples. The paper is quite well described with some minor fix needed.

1.       I did not see Table 2 in the script. It is missing

2.       If the ThromboFix/PBS with no replacement is your control protocol, what other control set are you using to generate data in Fig 1? Is it fresh sample at 0 hr? They are confusing, consider changing it to fresh sample instead?

3.       I would think plotting Fig 1a and Fig 2 together give the paper a better view of the set up that was described in Table 1.

4.       The naming is really confusing. Is P-PRP in pure water? In line 168, pure water replacement was proposed to be the best, but I can’t seem to tell which figure you are drawing the conclusion form.

5.       Consider running your polyp quantification in PBS (no replacement), PBS (water replacement), Plasma (water replacement) side by side. In its current form, it is quite messy.

Author Response

  1. I did not see Table 2 in the script. It is missing.

Response: This is my careless mistake. This is “Table 1.”

  1. If the ThromboFix/PBS with no replacement is your control protocol, what other control set are you using to generate data in Fig 1? Is it fresh sample at 0 hr? They are confusing, consider changing it to fresh sample instead?

Response: In the preliminary study, we tested platelet suspensions that were fixed with ThromboFix for 24 h, subsequently resuspended in pure water, and preserved at 4 °C. The data obtained in this case were similar to those of thromboFix-fixed platelet suspensions, at least within the initial 7 days. Thus, we considered the advantage of choosing a refrigerator rather than a freezer for preservation. Unfortunately, we could not find any advantage and thus decided to exclude the validation of this case. Instead, we concentrated on preservation at -80 °C.

Regarding the “fresh sample” you have indicated, we may not correctly understand what you mean by this expression. In all cases shown in this study, platelets were once fixed for at least 20 h at 4 °C (Line 99). Therefore, the time point at the end of fixation was defined as 0 h.

However, if you indicated “fresh sample” as a non-fixed sample, we could not obtain reproducible data from non-fixed samples.

  1. I would think plotting Fig 1a and Fig 2 together give the paper a better view of the set up that was described in Table 1.

Response: We agree with your opinion. If the three groups are compared equally, the idea is a good option. However, we aimed to show not only a comparison but also the practical issues of the current protocol. Thus, we isolated the data obtained from the current protocol in Figure 1. This is related to the purpose of this study (Lines 52-55).

  1. The naming is really confusing. Is P-PRP in pure water? In line 168, pure water replacement was proposed to be the best, but I can’t seem to tell which figure you are drawing the conclusion form.

Response: Please see Figure 2 (a vs. b). The time-course reduction of polyP levels in platelets suspended in PBS was less than 10% within 6 weeks. Compared to these data, the time-course reduction of polyP levels in platelets suspended in the plasma (P-PRP) exceeded 15% at 6 weeks. Therefore, we concluded that platelet suspension was better than P-PRP suspension. However, if the platelet count is lower and the preservation period is not so long, P-PRP suspension may be better owing to the minimization of platelet loss by centrifugation for the preparation of platelet suspension in PBS.

  1. Consider running your polyp quantification in PBS (no replacement), PBS (water replacement), Plasma (water replacement) side by side. In its current form, it is quite messy.

Response: We see how messy you felt. However, in a preliminary study, we tested many combinations and found that only these three combinations were comparable. We believe that researchers who are directly involved in the determination of DAPI-based polyP quantification will understand the meaning of these messy combinations.

Reviewer 4 Report

This paper describes the “Optimized protocol for preservation of human platelet samples for fluorometric polyphosphate quantification”. In general, the manuscript is well presented and the results are well interpreted. Overall, the manuscript is acceptable for publication.

Author Response

This paper describes the “Optimized protocol for preservation of human platelet samples for fluorometric polyphosphate quantification”. In general, the manuscript is well presented and the results are well interpreted. Overall, the manuscript is acceptable for publication.

Response: Thank you for your positive evaluation. We hope that this major revision, requested by some reviewers and the editor, will improve the article.

Round 2

Reviewer 2 Report

Although the authors responded satisfactorily to my general comments 2 and 3 of the report, the answer to comments in section 1 was not satisfactory. I still found the method used flawed and the paper should be rejected.

Author Response

Thank you for your comments.

Reviewer 3 Report

The comments were addressed, no more questions 

Author Response

Thank you for your comments.